# Context-PIPs: Persistent Independent Particles Demands Spatial Context Features

**Weikang Bian**[1,2][*] **Zhaoyang Huang**[1][*] **Xiaoyu Shi**[1]
**Yitong Dong**[3] **Yijin Li**[3] **Hongsheng Li**[1,2]
[1]CUHK MMLab  [2]Centre for Perceptual and Interactive Intelligence  [3]Zhejiang University
wkbian@outlook.com, drinkingcoder@link.cuhk.edu.hk, hsli@ee.cuhk.edu.hk

## Abstract

We tackle the problem of Persistent Independent Particles (PIPs), also called Tracking Any Point (TAP), in videos, which specifically aims at estimating persistent long-term trajectories of query points in videos. Previous methods attempted to estimate these trajectories independently to incorporate longer image sequences, therefore, ignoring the potential benefits of incorporating spatial context features. We argue that independent video point tracking also demands spatial context features. To this end, we propose a novel framework Context-PIPs, which effectively improves point trajectory accuracy by aggregating spatial context features in videos. Context-PIPs contains two main modules: 1) a SOurse Feature Enhancement (SOFE) module, and 2) a TArget Feature Aggregation (TAFA) module. Context-PIPs significantly improves PIPs all-sided, reducing 11.4% Average Trajectory Error of Occluded Points (ATE-Occ) on CroHD and increasing 11.8% Average Percentage of Correct Keypoint (A-PCK) on TAP-Vid-Kinetics. Demos are available at `https://wkbian.github.io/Projects/Context-PIPs/`.

## 1 Introduction

Video particles are a set of sparse point trajectories in a video that originate from the first frame (the source image) and move across the following frames, which are regarded as the target images. In contrast to optical flow estimation that computes pixel-wise correspondences between a pair of adjacent video frames, Persistent Independent Particles (PIPs) [12] or Tracking Any Point (TAP) [5] is interested in tracking the points in the follow-up frames that correspond to the original query points even when they are occluded in some frames. Video particles provide long-term motion information for videos and can support various downstream tasks, such as video editing [16] and Structure-from-Motion [48].

Long-range temporal information is essential for video particles especially when the particles are occluded because the positions of the occluded particles can be inferred from the previous and subsequent frames where they are visible. However, simultaneously encoding long image sequences brings larger computational and memory costs. Previous methods [12, 5] learn to track individual points independently because dense video particles are unnecessary in most scenarios. Inspired by optical flow estimation from visual similarities, they learn to predict point trajectories from the similarities between the query point and the subsequent target images. Specifically, given a query point at the source image, PIPs encodes $T$ feature maps from $T$ consecutive target images and builds a $T \times H \times W$ correlation volume by computing the feature similarity between the feature of the query point and the feature maps. The $T$ particle positions are iteratively refined with the 3D correlation volume through a shared MLP-Mixer [40]. In other words, PIPs trades the spatial context features of

---

[*]Weikang Bian and Zhaoyang Huang assert equal contributions.

37th Conference on Neural Information Processing Systems (NeurIPS 2023), Vancouver, Canada.

the particle for longer temporal feature encoding. PIPs achieves great performance on the DAVIS dataset, which contains large movement particles and weak texture images (e.g., fast-moving dogs and black bears).

We argue that independent point tracking still demands spatial context features. Intuitively, although PIPs only accounts for specified query points, spatial context features around them provide informative cues for point trajectory refinement. For example, video particles on the same objects always share similar motions over time. In some video frames where the target particles are occluded, their surrounding particles may be visible and provide guidance for the position estimation of the target particles. However, PIPs only takes correlations and features belonging to the particles while ignoring abundant spatial context features around them. In this work, we propose tracking particles with Context (Context-PIPs) to improve independent point tracking with spatial context features. Context-PIPs contains two key modules for better point trajectory refinement: 1) a source feature enhancement (SOFE) module that learns to adopt more spatial context features in the source image and builds a guidance correlation volume, and 2) a target feature aggregation (TAFA) module that aggregates spatial context features in the target image guided by the correlation information.

In the source image, points that possess similar appearances are supposed to move in similar trajectories in subsequent frames. Such an assumption has also been used in GMA [21] for optical flow estimation. Given a query point, SOFE computes the correlation between the query point and the source image feature map, which is its self-similarity map. With the guidance of the correlation (self-similarity), SOFE predicts $M$ offsets centered from the query point, and samples at the corresponding $M$ auxiliary points to collect source context features. During the iterative point trajectory refinement, the correlation information between the $M$ auxiliary features and $T$ target feature maps is injected into the MLP-Mixer, which provides strong guidance and shows evident performance improvement.

Existing methods for optical flow and video particle estimation ignore features in target images for iterative refinement. To better utilize the context of target images, in each iteration, our TAFA module collects target features surrounding the previous iteration's resulting movements. TAFA for the first time shows that context features in target images also benefit correspondence estimation and further improve the point tracking accuracy.

Our contributions can be summarized as threefold: 1) We propose a novel framework, Context-PIPs, to improve independent video particle tracking with context features from both source and target features. Context-PIPs ranks 1st on the four benchmarks and shows clear performance superiority. 2) We design a novel SOurce Feature Enhancement (SOFE) module that builds a guidance correlation volume with spatial context features in the source image, and 3) a novel TArget Feature Aggregation (TAFA) module that extracts context features from target images.

## 2   Related Work

**Optical Flow.** Optical flow estimates the dense displacement field between image pairs and has traditionally been modeled as an optimization problem that maximizes the visual similarity between image pairs with regularizations [13, 1, 2, 34]. It serves as the core module of many downstream applications, such as Simultaneously Localization And Mapping (SLAM) [8, 47, 27, 43, 9], video synthesis [46, 14, 15, 42, 41], video restoration [24, 23], etc. Since FlowNet [6, 20], learning optical flow with neural networks presents superior performance over traditional optimization-based methods and is fast progressing with more training data obtained by the renderer and better network architecture [6, 20, 28, 35, 36, 18, 19, 45]. In recent years, iterative refining flow with all-pairs correlation volume presents the best performance. The most successful network designs are RAFT [39] and FlowFormer [17, 32], which achieves state-of-the-art accuracy.

Typical optical flow estimation only takes image pairs but longer image sequences can provide more information that benefits optical flow estimation. Kalman filter [4, 7] had been adopted in dealing with the temporal dynamics of motion and estimating multi-frame optical flow. Recent learning-based methods also attempted to exploit multi-frame information and perform multi-frame optical flow estimation. PWC-Fusion [29] is the first method that learns to estimate optical flow from multiple images. However, it only fuses information from previous frames in U-Net and improves little performance. The "warm start" technique [39, 33, 37] that wrapped the previous flow to initialize the next flow is firstly proposed in RAFT and shows clear accuracy increasing. VideoFlow [31] takes multi-frame cues better, iteratively fusing multi-frame information in a three-frame and five-

frame structure, which reveals that longer temporal information benefits pixel tracking. Recently, researchers [25, 10] also tried to learn to estimate optical flow with event cameras.

**Tracking Any Point.** Optical flow methods merely focus on tracking points between image pairs but ignore tracking points across multiple consecutive frames, which is still challenging. Harley *et al.* [12] studied pixel tracking in the video as a long-range motion estimation problem inspired by Particle Video [30]. They propose a new dataset FlyingThings++ based on FlyingThings [26] for training and Persistent Independent Particles (PIPs) to learn to track single points in consecutive frames with fixed lengths. Doersch *et al.* [5] is a parallel work, which formalized the problem as tracking any point(TAP). They also propose a new dataset Kubric [5] for training and a network TAP-Net to learn point tracking. Moreover, they provide the real video benchmarks that are labeled by humans, TAP-Vid-DAVIS [5] and TAP-Vid-Kinetics [5], for evaluation. PIPs and TAP solve the video particle tracking problem in a similar manner, i.e., recurrently refining multi-frame point trajectory via correlation maps. In this paper, our Context-PIPs follows the training paradigm of PIPs and improves the network architecture design of PIPs. We also take the TAP-Vid-DAVIS and TAP-Vid-Kinetics benchmarks from TAP-Net for evaluation.

## 3 Method

In contrast to optical flow methods [39, 17] that track dense pixel movement between an image pair, the problem of point tracking takes $T$ consecutive RGB images with a single query point $\mathbf{x}_{src} \in \mathbb{R}^2$ at the first frame as input, and estimates $T$ coordinates $\mathbf{X} = \{\mathbf{x}_0, \mathbf{x}_1, \ldots, \mathbf{x}_{T-1}\}$ at the video frames where every $\mathbf{x}_t$ indicates the point's corresponding location at time $t$. Persistent Independent Particles (PIPs) [12] is the state-of-the-art network architecture for TAP. It iteratively refines the point trajectory by encoding correlation information that measures visual similarities between the query point and the $T$ video frames. The query points to be tracked are easily lost when the network only looks at them and ignores spatial context features. We propose a novel framework Context-PIPs (Fig. 1) that improves PIPs with a SOurce Feature Enhancement (SOFE) module and a TArget Feature Aggregation (TAFA) module. In this section, we first briefly review PIPs and then elaborate on our Context-PIPs.

### 3.1 A Brief Revisit of PIPs

PIPs [12] processes $T$ video frames containing $N$ independent query points simultaneously and then extends the point trajectories to more video frames via chaining rules [12]. Given a source frame with a query point $\mathbf{x}_{src} \in \mathbb{R}^2$ and $T-1$ follow-up target video frames, PIPs first extracts their feature maps $\mathbf{I}_0, \mathbf{I}_1, \ldots, \mathbf{I}_{T-1} \in \mathbb{R}^{C \times H \times W}$ through a shallow convolutional neural network and bilinearly samples to obtain the source point feature $\mathbf{f}_{src} = \mathbf{I}_0(\mathbf{x}_{src})$ from the first feature map at the query point $\mathbf{x}_{src}$. $C, H, W$ are the feature map channels, height, and width. Inspired by RAFT [39], PIPs initializes the point trajectory and point visual features at each frame with the same $\mathbf{x}_{src}$ and $\mathbf{f}_{src}$:

$$\begin{aligned} \mathbf{X}^0 &= \{\mathbf{x}_0^0, \mathbf{x}_1^0, \ldots, \mathbf{x}_{T-1}^0 | \mathbf{x}_t^0 = \mathbf{x}_{src}, t = 0, \ldots, t = T - 1\}, \\ \mathbf{F}^0 &= \{\mathbf{f}_0^0, \mathbf{f}_1^0, \ldots, \mathbf{f}_{T-1}^0 | \mathbf{f}_t^0 = \mathbf{f}_{src}, t = 0, \ldots, t = T - 1\}, \end{aligned} \tag{1}$$

and iteratively refines them via correlation information. $\mathbf{x}_t^k$ and $\mathbf{f}_t^k$ respectively denote the point trajectory and point features in the $t$-th frame and $k$-th iteration. Intuitively, the point features store the visual feature at the currently estimated query point location in all the $T$ frames.

Specifically, in each iteration $k$, PIPs constructs multi-scale correlation maps [39] between the guidance feature $\{\mathbf{f}_t^k\}_{t=0}^{T-1}$ and the target feature maps $\{\mathbf{I}_t^k\}_{t=0}^{T-1}$, which constitutes $T$ correlation maps $\mathbf{C}^k = \{\mathbf{c}_0^k, \mathbf{c}_1^k, \ldots, \mathbf{c}_{T-1}^k\}$ of size $T \times H \times W$, and crops correlation information inside the windows centered at the point trajectory: $\mathbf{C}^k(\mathbf{X}^k) = \{\mathbf{c}_0^k(\mathbf{x}_0^k), \mathbf{c}_1^k(\mathbf{x}_1^k), \ldots, \mathbf{c}_{T-1}^k(\mathbf{x}_{T-1}^k)\}$, where $\mathbf{c}_t^k(\mathbf{x}_t^k) \in \mathbb{R}^{D \times D}$ denotes that we crop $D \times D$ correlations from $\mathbf{c}_t^k$ inside the window centered at $\mathbf{x}_t^k$. The point features $\mathbf{F}^k$, point locations $\mathbf{X}^k$, and the local correlation information $\mathbf{C}^k(\mathbf{X}^k)$ are fed into a standard 12-layer MLP-Mixer that produces $\Delta \mathbf{F}$ and $\Delta \mathbf{X}$ to update the point feature and the point trajectory:

$$\begin{aligned} \Delta \mathbf{F}, \ \Delta \mathbf{X} &= \text{MLPMixer}(\mathbf{F}^k, \mathbf{C}^k(\mathbf{X}^k), \text{Enc}(\mathbf{X}^k - \mathbf{x}_{src})), \\ \mathbf{F}^{k+1} &= \mathbf{F}^k + \Delta \mathbf{F}, \ \mathbf{X}^{k+1} = \mathbf{X}^k + \Delta \mathbf{X}. \end{aligned} \tag{2}$$

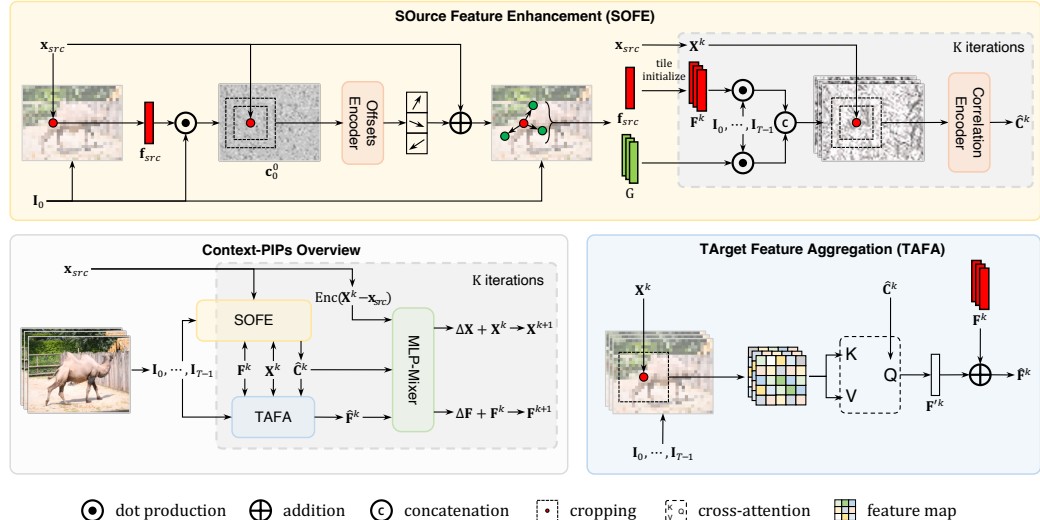

Figure 1: Overview of Context-PIPs Pipeline. Our Context-PIPs improves PIPs [12] with SOurce Feature Enhancement (SOFE) and TArget Feature Aggregation (TAFA). PIPs iteratively refines the point trajectory $\mathbf{X}^k$ with an MLP-Mixer with the current point trajectory $\mathbf{X}^k$, the correlation features $\mathbf{C}^k$, and the point features $\mathbf{F}^k$. SOFE and TAFA respectively improves the correlation features and point features, denoted as $\hat{\mathbf{C}}^k$ and $\hat{\mathbf{F}}^k$.

PIPs iterates $K$ times for updates and the point trajectory in the last iteration $\mathbf{X}^K$ is the output.

PIPs achieves state-of-the-art accuracy on point tracking by utilizing longer temporal features. However, the previous method ignores informative spatial context features which are beneficial to achieve more accurate point tracking. Context-PIPs keeps all modules in PIPs and is specifically designed to enhance the correlation information $\mathbf{C}^k$ and point features $\mathbf{F}^k$ as $\hat{\mathbf{C}}^k$ and $\hat{\mathbf{F}}^k$ with the proposed SOFE and TAFA.

### 3.2 Source Feature Enhancement

Given the query point $\mathbf{x}_{src}$ and feature map $\mathbf{I}_0$ of the source image, PIPs simply samples a source feature $\mathbf{f}_{src}$ at the query point location to obtain the point visual features $\mathbf{F}^k$. Although the point features are updated via the iterative refinement, its perceptive field is limited to the single point, easily compromised in harsh scenarios. The correlation maps $\mathbf{C}^k$ in the $k$-th iteration provide vague information when the query point is in a less textured area. Moreover, the correlation map $\mathbf{c}_t^k$ at timestamp $k$ is ineffective once the particle is occluded at the $t$-th frame. To enhance the source feature, as shown in Fig. 1, we propose SOurce Feature Enhancement (SOFE) that accepts spatial context features in the source image as auxiliary features to guide the point trajectory refinement. The MLP-Mixer can infer the point locations via the auxiliary features even when the points are occluded or on less textured regions, which improves the point tracking accuracy and robustness.

Directly adopting all features in the source image brings large computational costs. SOFE learns to sample a small number of auxiliary features to enhance the source feature. Specifically, SOFE improves the point features in three steps. Firstly, SOFE learns to predict $M$ offsets $\delta\mathbf{x}_0, \delta\mathbf{x}_1, \ldots, \delta\mathbf{x}_{M-1} \in \mathbb{R}^2$ with an MLP-based sampler to sample $M$ auxiliary features $G = \{\mathbf{g}_0, \mathbf{g}_1, \ldots, \mathbf{g}_{M-1} | \mathbf{g}_m = \mathbf{I}_0(\mathbf{x}_{src} + \delta\mathbf{x}_m)\}$ around the query point $\mathbf{x}_{src}$ in the source image. Motivated by GMA that aggregates pixel flows from pixels that are likely to belong to the same object through self-similarity, our proposed sampler also learns the locations of the auxiliary features based on local self-similarities $\mathbf{c}_0^0(\mathbf{x}_{src})$ which store the correlations cropped from the first frame at the query point location. Secondly, we construct the correlation map $\mathbf{c}'_{m,t} =< \mathbf{g}_m, \mathbf{I}_t > \in \mathbb{R}^{H \times W}$ that measure visual similarities of the $m$-th auxiliary feature and the $t$-th frame feature map. $\hat{\mathbf{c}}_{m,t}$ provides additional correlation information to guide the iterative point trajectory refinement. In each iteration $k$, we crop the additional correlation information $\mathbf{c}'_m(\mathbf{x}_t^k)$ according to the point locations

$\mathbf{x}_t^k$ and concatenate them with the original point correlation information $\mathbf{c}_t^k(\mathbf{x}_t^k)$, where $\mathbf{c}_m'(\mathbf{x}_t^k)$ denotes the same cropping operation as $\mathbf{c}_t^k(\mathbf{x}_t^k)$. Finally, for each frame $t$, we reduce the augmented correlations to a correlation feature vector $\hat{\mathbf{c}}_t$ of length 196 through a correlation encoder CorrEnc.

$$\hat{\mathbf{c}}_t^k = \mathrm{CorrEnc}(\mathrm{Concat}(\mathbf{c}_0'(\mathbf{x}_t^k), \mathbf{c}_1'(\mathbf{x}_t^k), \ldots, \mathbf{c}_{M-1}'(\mathbf{x}_t^k), \mathbf{c}_t^k(\mathbf{x}_t^k))), \tag{3}$$

and inject $\hat{\mathbf{C}}^k = \{\hat{\mathbf{c}}_0^k, \hat{\mathbf{c}}_1^k, \ldots, \hat{\mathbf{c}}_{T-1}^k, \}$ into the MLP-Mixer. Compared with PIPs that only adopts $\mathbf{c}_t^k(\mathbf{x}_t^k))$, SOFE provides more informative correlations to the MLP-Mixer with spatial context features but does not increase its parameters and computations. The additional auxiliary features from the self-similarity map of the source image enhance the original source point features and significantly improves tracking accuracy.

### 3.3  Target Feature Aggregation

Inspired by existing optical flow methods, PIPs iteratively refines the point trajectory with correlation information and context features and also iteratively updates the point visual feature $\mathbf{F}^{k+1} = \mathbf{F}^k + \Delta\mathbf{F}$ after initializing them with the source feature, which presents benefits for point trajectory refinement. We observe that the input for supporting the point feature updating comes from the correlations $\mathbf{C}^k$ only. However, such correlations $\mathbf{C}^k$ are calculated as only cosine distances between the source point visual feature $\mathbf{F}^k$ and the target features around currently estimated point locations $\mathbf{X}^k$, which provide limited information on how to conduct visual feature updating. Can we better guide the point feature update with context features in target images?

We, therefore, propose TArget Feature Aggregation (TAFA) to augment point features with target image features nearby the point trajectory. Specifically, for each target frame $t$, a patch of shape $D \times D$ cropped from the corresponding target feature map $\mathbf{I}_t$ centered at $\mathbf{x}_t^k$ to generate key and value. The augmented correlation features $\hat{\mathbf{C}}$ in Eq. 3 encode abundant visual similarities. Therefore, we generate a query from it to extract target context features and adopt cross-attention with relative positional encoding to obtain the target context feature $\mathbf{f}'^k_t$, which is added to the original source point feature $\hat{\mathbf{f}}_t^k = \mathbf{f}_t^k + \mathbf{f}'^k_t$. Finally, such augmented point features $\hat{\mathbf{F}}^k = \{\hat{\mathbf{f}}_0^k, \hat{\mathbf{f}}_1^k, \ldots, \hat{\mathbf{f}}_{T-1}^k\}$ are injected into the MLP-Mixer. Similar to our proposed SOFE, TAFA also keeps the same parameters and computations of MLP-Mixer as PIPs while providing additional target context features and further improving PIPs. Although context features in the source image are used since RAFT [39], no previous methods adopt context features from target images. TAFA for the first time validates that target images also contain critical context features that benefit point movement refinement. SOFE improves PIPs with auxiliary features in the source image while TAFA absorbs more target image features. Equipping SOFE and TAFA to PIPs constitutes our final model, Context-PIPs.

### 3.4  Loss Functions

We compute the L1 distance between $\mathbf{X}^k$ computed in iteration $k$ and the ground truth $\mathbf{X}_{gt}$ and constrain with exponentially increasing weights $\gamma = 0.8$.

$$\mathcal{L}_{TAP} = \sum_{k=1}^{K} \gamma^{K-k} ||\mathbf{X}^k - \mathbf{X}_{gt}||_1 \tag{4}$$

In addition, we will predict the visibility/occlusion $\mathbf{V}$ by a linear layer according to the $\hat{\mathbf{F}}^K$ obtained by the iterative update. And the cross entropy loss is used to supervise $\mathbf{V}$ with the ground truth $\mathbf{V}_{gt}$.

$$\mathcal{L}_{Vis} = \mathbf{V}_{gt} \log \mathbf{V} + (1 - \mathbf{V}_{gt}) \log(1 - \mathbf{V}). \tag{5}$$

The final loss is the weighted sum of the two losses:

$$\mathcal{L}_{total} = w_1 \mathcal{L}_{TAP} + w_2 \mathcal{L}_{Vis}. \tag{6}$$

We use $w_1 = 1$ and $w_2 = 10$ during training.

Table 1: Experiments on FlyingThings++, CroHD, TAP-Vid-Davis, and TAP-Vid-Kinetics (first). Vis. and Occ. denotes ATE-Vis. and ATE-Occ.

| Method | FlyingThings++ | | CroHD | | DAVIS (first) | | Kinetics (first) | |
|---|---|---|---|---|---|---|---|---|
| | Vis. | Occ. | Vis. | Occ. | AJ | A-PCK | AJ | A-PCK |
| DINO | 43.05 | 76.25 | 22.50 | 26.06 | - | - | - | - |
| RAFT | 16.75 | 43.21 | 7.99 | 13.16 | 27.1 | 42.1 | 35.1 | 49.7 |
| TAP-Net | - | - | 17.00 | 20.86 | 33.0 | 48.6 | 38.5 | 54.4 |
| PIPs (Paper) | 15.54 | 36.67 | 5.16 | 7.56 | - | - | - | - |
| PIPs (Re-imp.) | 7.40 | 24.40 | 4.73 | 7.97 | 39.2 | 55.1 | 33.4 | 51.0 |
| Context-PIPs (Ours) | **6.44** | **22.22** | **4.28** | **7.06** | **42.7** | **60.3** | **40.2** | **57.0** |

# 4 Experiments

We evaluate our Context-PIPs on four benchmarks: FlyingThings++ [12], CroHD [38], TAP-Vid-DAVIS, and TAP-Vid-Kinetics [5]. Following PIPs [12], we train Context-PIPs on Flyingthings++ only and evaluate it on other benchmarks without finetuning. Context-PIPs achieves state-of-the-art performance on all benchmarks and significantly improves PIPs. Moreover, we show that by utilizing spatial context features in Context-PIPs, we achieve on-par performance with PIPs when using only 40.2% of its parameters.

**Datasets** Flyingthings++ is a synthetic dataset based on Flyingthings3D [26], which contains 8-frame trajectories with occlusion. The video resolution is $384 \times 512$ for both training and evaluation. Crowd of Heads Dataset (CroHD) is a high-resolution crowd head tracking dataset. Following PIPs, the RAFT [39], PIPs, and our Context-PIPs are evaluated at $768 \times 1280$ resolution. DINO [3] and TAP-Net [5] are respectively evaluated at $512 \times 768$ and $256 \times 256$ resolution. TAP-Vid-DAVIS and TAP-Vid-Kinetics are two evaluation datasets in the TAP-Vid benchmark, both of which consist of real-world videos with accurate human annotations for point tracking. Note that TAP-Vid provides two distinct query sampling strategies, i.e., first and strided. The "first" sampling contains only the initial visible query points until the last frame, while the "strided" sampling is to sample all visible query points in every 5 frames. RAFT, TAP-Net, PIPs, and our Context-PIPs are evaluated with two different sampling strategies respectively at $256 \times 256$ resolution. While Kubric-VFS-like [44, 11] and COTR [22] are only evaluated with the "first" sampling strategies at the same resolution.

**Experiment Setup** We use the average trajectory error (ATE) metric [12] for evaluation on FlyingThings++ and CroHD. ATE measures the average L2 distances between the coordinates of all predicted points in the trajectories and the corresponding ground truth coordinates. According to the ground truth visibility of the points, we calculate ATEs for all visible points and occluded points separately. i.e., ATE-Vis. and ATE-Occ. We use the average Jaccard (AJ) [12], the average percentage of correct keypoint (A-PCK) metrics for TAP-Vid-DAVIS and TAP-Vid-Kinetics. The Jaccard metric measures the ratio of "true positive" visible points within a given threshold. Average Jaccard (AJ) averages Jaccard across the different thresholds. PCK measures the percentage of the predicted coordinates whose L2 distances from the ground truth are smaller than a given threshold. A-PCK refers to calculating PCK according to multiple different thresholds and then taking the average. We set the thresholds as 1, 2, 4, 8, and 16.

**Implementation Details** We train our Context-PIPs with a batch size of 4 and 100,000 steps on Flyingthings++ with horizontal and vertical flipping. We use the one-cycle learning rate scheduler. The highest learning rate is set as $5 \times 10^{-4}$. During training, we set the convolution stride to 8 and the resolution of the input RGB images to $384 \times 512$, and randomly sample $N = 128$ visible query points for supervision. To limit the length of the input videos, we set $T = 8$ and apply the trajectory linking mechanism [12] at test time, similarly to PIPs. To align to the PIPs paper, the PIPs compared in Tab. 1 and Tab. 2 is trained with $K = 6$. In the ablation study, all models are trained with $K = 4$ while tested with $K = 6$ for a fair comparison.

## 4.1 Quantitative Comparison

**FlyingThings++ and CroHD** As shown in Tab. 1, Context-PIPs ranks 1st on all metrics and presents significant performance superiority compared with previous methods. Specifically, Context-PIPs

Table 2: Experiments on TAP-Vid-DAVIS(strided) and TAP-Vid-Kinetics(strided).

| Method | TAP-DAVIS(strided) | | TAP-Kinetics(strided) | |
|---|---|---|---|---|
| | AJ | A-PCK | AJ | A-PCK |
| RAFT | 30.0 | 46.3 | 34.5 | 52.5 |
| Kubric-VFS-Like | 33.1 | 48.5 | 40.5 | 59.0 |
| COTR | 35.4 | 51.3 | 19.0 | 38.8 |
| TAP-Net | 38.4 | 53.1 | 46.6 | 60.9 |
| PIPs(Re-imp.) | 45.2 | 59.8 | 42.9 | 58.3 |
| Context-PIPs (Ours) | **48.9** | **64.0** | **49.8** | **64.3** |

Table 3: Occlusion Accuracy on TAP-Vid-DAVIS and TAP-Vid-Kinetics.

| Method | First | | Strided | |
|---|---|---|---|---|
| | TAP-DAVIS | TAP-Kinetics | TAP-DAVIS | TAP-Kinetics |
| TAP-Net | 78.8 | 80.6 | 82.3 | 85.0 |
| PIPs (Re-imp.) | 79.3 | 77.0 | 82.9 | 81.5 |
| Context-PIPs (Ours) | 79.5 | 79.8 | 83.4 | 83.3 |

achieves 7.06 ATE-Occ and 4.28 ATE-Vis on the CroHD dataset, 11.4% and 9.5% error reductions from PIPs, the runner-up. On the FlyingThings++ dataset, our Context-PIPs decreases the ATE-Vis and ATE-Occ by 0.96 and 2.18, respectively.

**TAP-Vid-DAVIS and TAP-Vid-Kinetics (first)** A-PCK, the average percentage of correct key points, is the core metric. Context-PIPs ranks 1st in terms of A-PCK on both benchmarks. Specifically, Context-PIPs outperforms TAP-Net by 24.1% on the TAP-Vid-DAVIS benchmark and improves PIPs by 11.8% on the TAP-Vid-Kinetics benchmarks. Context-PIPs also achieves state-of-the-art occupancy accuracy on both datasets.

**TAP-Vid-DAVIS and TAP-Vid-Kinetics (strided)** Tab. 2 compares methods in the "strided" sampling setting. Our Context-PIPs also achieves the best performance on both AJ and A-PCK metrics for the two datasets. Context-PIPs effectively improves PIPs' occupancy accuracy and presents the best performance on the TAP-Vid-Davis dataset.

## 4.2 Qualitative Comparison

We visualize the trajectories estimated by TAP-Net, PIPs, and our Context-PIPs respectively in Fig 2 to qualitatively demonstrate the superior performance of our method. By incorporating additional spatial context features for point tracking, Context-PIPs surpasses the compared methods in accuracy and robustness. Specifically, the first row shows the case of large-scale variation. The trajectory predicted by TAP-Net deviates considerably from the ground truth. TAP-Net also outputs jittery predictions when the query pixel is on the texture-less area as shown in the second row. Our Context-PIPs generates more accurate results than PIPs in these two hard cases. Furthermore, as depicted in the third row, PIPs struggles to distinguish the front wheel and the rear wheel due to the changing lighting conditions. However, our Context-PIPs achieves consistent tracking, thanks to the rich context information brought by the SOFE and TAFA modules.

## 4.3 Efficiency Analysis

We train our Context-PIPs and PIPs with different MLP-Mixer depths, i.e., the number of layers in the MLP-Mixer, to show the prominent efficiency and effectiveness of our proposed Context-PIPs. Context-PIPs improves PIPs with SOFE and TAFA, which introduce minor additional parameters and time costs. We show that the accuracy benefits do not come from the increased parameters trivially. As displayed in Tab. 4, we increase the MLP-Mixer depth to 16, which significantly increases the parameters but does not bring performance gain. We also decrease the MLP-Mixer depth in our Context-PIPs. Even with only a 3-layer MLP-Mixer, Context-PIPs achieves better performance than the best PIPs (MLP-Mixer depth=12). Context-PIPs outperforms PIPs with only 40.2% parameters. Moreover, evaluated by the pytorch-OpCounter [49], PIPs consumes 287.5G FLOPS while Context-

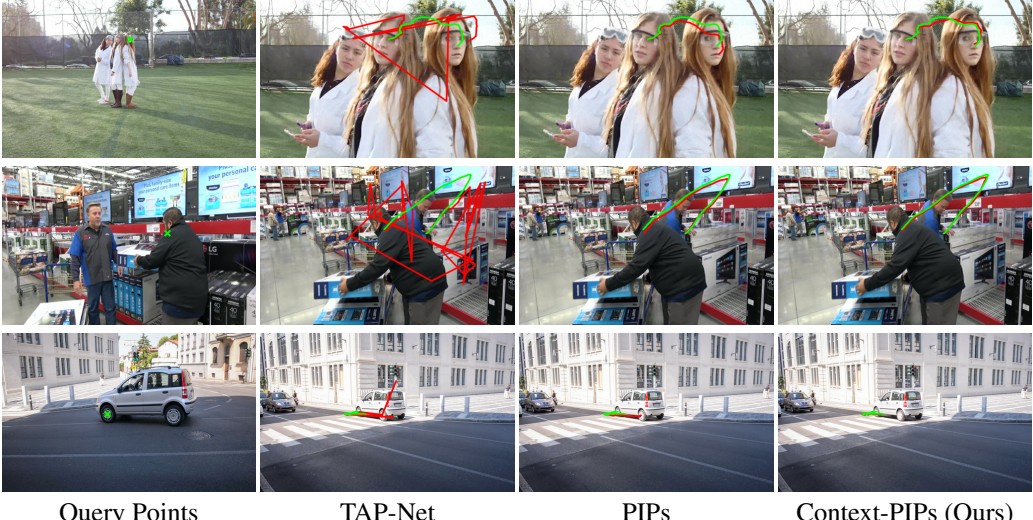

| Query Points | TAP-Net | PIPs | Context-PIPs (Ours) |

Figure 2: Qualitative comparison. In the leftmost column, the green crosses mark the query points and the image is the starting frame. The right three columns show the results of TAP-Net, PIPs, and Context-PIPs. The red and green lines illustrate the predicted and ground truth trajectories.

Table 4: Efficiency analysis for PIPs and Context-PIPs with different MLP-Mixer depth.

| Method | MLP-Mixer Depth | Param.(M) | Flyingthings++ | | CroHD | |
|---|---|---|---|---|---|---|
| | | | vis | occ | vis | occ |
| PIPs | 6 | 16.06 | 8.00 | 25.30 | 5.04 | 8.15 |
| | 12 | 28.67 | 7.70 | 24.70 | 4.98 | 8.20 |
| | 16 | 37.09 | 7.69 | 24.71 | 4.84 | 8.07 |
| Context-PIPs (Ours) | 3 | 11.54 | 7.37 | 24.19 | 4.54 | 7.79 |
| | 6 | 17.84 | 6.94 | 22.56 | 4.37 | 7.05 |
| | 12 | 30.46 | 6.60 | 22.11 | 4.30 | 6.73 |

PIPs only consumes 216.4G FLOPS, saving 24.7% computation resources. These numbers reveal the high memory and computation efficiency of our proposed Context-PIPs.

## 4.4 Ablation Study on Modules

We conduct a module ablation study on the proposed modules as presented in Tab. 5. The errors of Context-PIPs consistently decrease when we sequentially add the SOFE and TAFA modules, which reveals the effectiveness of SOFE and TAFA. To demonstrate the necessity of the cross-attention mechanism used in TAFA, we attempt to predict a matrix of weights corresponding to the feature map shaped with $r_a$ and directly weigh and sum the features to get $\delta F$. Cross-attention performs better than prediction.

## 4.5 Ablation Study on Parameters

We conduct a series of ablation experiments (Tab. 6) to demonstrate the significance of each module and explain the rationality of the settings. All ablation experiments are trained on Flyingthings++. Starting from the PIPs baseline, we first add the SOFE module and explore the two related hyperparameters, i.e., the correlation radius $r_c$ and the number of samples $M$. Then, we further add the TAFA module and also adjust the attention window radius $r_a$. We additionally conduct a comparison between the prediction and attention mechanisms in TAFA. In the below experiments, we set $N = 64$, learning rate as $3 \times 10^{-4}$, and train for $20,000$ steps. Below we describe the details.

**Correlation Radius in SOFE** We crop a multi-scale correlation of size $(2r_c + 1) \times (2r_c + 1)$ from the first correlation map to predict the auxiliary feature offsets in SOFE. The correlation radius $r_c$

Table 5: Modules ablation study.

| Method | Experiment | Flyingthings++ | | CroHD | |
|---|---|---|---|---|---|
| | | vis | occ | vis | occ |
| PIPs | Baseline | 7.40 | 24.4 | 4.73 | 7.97 |
| Context-PIPs | +SOFE | 6.91 | 22.64 | 4.36 | 7.15 |
| | +SOFE+TAFA (attention) | 6.60 | 22.11 | 4.30 | 6.73 |
| | +SOFE+TAFA (prediction) | 7.15 | 23.33 | 4.34 | 7.27 |

Table 6: Ablation study. We add one component at a time on top of the baseline to obtain our Context-PIPs. $r_c$, $M$, and $r_a$ respectively denote the correlation radius and the number of predicted samples in SOFE, and the attention window radius in TAFA. Our final model uses $M = 9, r_c = 2, r_a = 3$ to achieve the best performance.

| Method | Experiment | Parameters | | | Flyingthings++ | | CroHD | |
|---|---|---|---|---|---|---|---|---|
| | | $M$ | $r_c$ | $r_a$ | vis | occ | vis | occ |
| PIPs | Baseline | - | - | - | 14.42 | 37.33 | 6.14 | 9.97 |
| +SOFE | Correlation Radius ($r_c$) | 3 | 1 | - | 13.60 | 36.17 | 6.40 | 10.30 |
| | | 3 | 2 | - | **13.02** | **35.60** | 6.48 | **9.69** |
| | | 3 | 3 | - | 13.07 | 35.74 | **6.23** | 10.09 |
| | | 3 | 4 | - | 13.75 | 37.01 | 6.64 | 10.20 |
| | Number of Samples ($M$) | 1 | 2 | - | 15.00 | 37.51 | 6.94 | 10.25 |
| | | 3 | 2 | - | 13.02 | 35.60 | 6.48 | 9.69 |
| | | 6 | 2 | - | 13.21 | 35.39 | 6.58 | 9.77 |
| | | 9 | 2 | - | **12.18** | **34.23** | **5.71** | **9.18** |
| | | 12 | 2 | - | 12.87 | 35.27 | 6.20 | 10.17 |
| +SOFE+TAFA | Attention Window ($r_a$) | 9 | 2 | 1 | 11.98 | 34.10 | 5.90 | 9.52 |
| | | 9 | 2 | 2 | 11.82 | 33.82 | 5.64 | 9.28 |
| | | 9 | 2 | 3 | **11.67** | **33.38** | **5.53** | 9.19 |
| | | 9 | 2 | 4 | 11.81 | 33.88 | 5.65 | 9.23 |
| | | 9 | 2 | 5 | 11.83 | 33.76 | 5.60 | **9.15** |

determines the cropping patch size. We fix $M = 3$, and gradually increase $r_c$ from 1 to 4. The model achieves the best performance when $r_c = 2$.

**Number of Samples in SOFE** SOFE learns to sample $M$ additional auxiliary features to enhance the source feature. Given $r_c = 2$, we continued to experiment with the different number of samples $M$. The model achieves the best performance on both Flyingthings++ and CroHD datasets when $M = 9$.

**Attention Radius in TAFA** TAFA aggregates target features surrounding the currently estimated corresponding point locations to enhance the context feature via cross-attention. The radius of the attention window $r_a$ determines how far the attention can reach. We gradually increase $r_a$ from 1 up to 5, and find that $r_a = 3$ performs best.

## 5  Conclusion

We have presented a novel framework Context-PIPs that improves PIPs with spatial context features, including a SOurce Feature Enhancement (SOFE) module and a TArget Feature Aggregation (TAFA) module. Experiments show that Context-PIPs achieves the best tracking accuracy on four benchmark datasets with significant superiority. This technology has broad applications in video editing and 3D reconstruction and other fields. **Limitations**. Following PIPs, Context-PIPs tracks points in videos with a sliding window. The target point cannot be re-identified when the point is lost. In our future work, we will explore how to re-identify the lost points when the points are visible again.
**Acknowlegement** This project is funded in part by National Key R&D Program of China Project 2022ZD0161100, by the Centre for Perceptual and Interactive Intelligence (CPII) Ltd under the Innovation and Technology Commission (ITC)'s InnoHK, by General Research Fund of Hong Kong RGC Project 14204021. Hongsheng Li is a PI of CPII under the InnoHK.

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

# Appendix

## A  More Implementation Details

**SOFE sampler**. While sampling auxiliary features, SOFE learns to predict offsets with an MLP-based sampler which consists of 5 linear layers interweaved with RELU activations. The local self-similarities $\mathbf{c}_0^0$ at location $\mathbf{x}_{src}$) would first be projected into 128 feature channels. A 3-layer feedforward network with $4 \times 128$ channels followed, outputting the feature in 128 feature channels. The final linear layer is used to predict offsets $M \times 2$ from the 128-channel features.

**SOFE CorrEnc**. SOFE reduces the augmented correlations $(M + 1) \times 196$ to a correlation feature vector $\hat{\mathbf{c}}_t^k$ through a correlation encoder $\mathrm{CorrEnc}$ which contains only 2 linear layers. The first linear layer reduces the feature channels to $4 \times 196$. After a RELU, the later linear layer further reduces the feature channels to 196, which is the $\hat{\mathbf{c}}_t^k$.

Table A1: Experiments on FlyingThings++ and CroHD datasets.

| Method | FlyingThings++ | | CroHD | |
|---|---|---|---|---|
| | ATE-Vis. | ATE-Occ. | ATE-Vis. | ATE-Occ. |
| TAP-Net | - | - | 17.00 | 20.86 |
| PIPs (Paper) | 15.54 | 36.67 | 5.16 | 7.56 |
| PIPs (Re-imp., $K = 4$) | 7.70 | 24.70 | 4.98 | 8.20 |
| PIPs (Re-imp., $K = 6$) | 7.40 | 24.40 | 4.73 | 7.97 |
| PIPs (Released) | **6.08** | **19.15** | 4.56 | 7.71 |
| Context-PIPs (Ours, $K = 4$) | 6.60 | 22.11 | 4.30 | **6.73** |
| Context-PIPs (Ours, $K = 6$) | 6.44 | 22.22 | **4.28** | 7.06 |

Table A2: Experiments on TAP-Vid-DAVIS (first), and TAP-Vid-Kinetics (first).

| Method | $K$ | MLP-Mixer | TAP-Vid-DAVIS (first) | | TAP-Vid-Kinetics (first) | |
|---|---|---|---|---|---|---|
| | | Depth | AJ | A-PCK | AJ | A-PCK |
| TAP-Net | - | - | 33.0 | 48.6 | 38.5 | 54.4 |
| PIPs (Re-imp.) | 4 | 6 | 34.4 | 51.6 | 28.8 | 48.6 |
| PIPs (Re-imp.) | 4 | 12 | 34.6 | 51.8 | 28.8 | 47.2 |
| PIPs (Re-imp.) | 4 | 16 | 35.3 | 52.5 | 29.9 | 48.5 |
| PIPs (Re-imp.) | 6 | 12 | 39.2 | 55.1 | 33.4 | 51.0 |
| PIPs(Released) | - | - | 38.5 | 55.4 | 30.1 | 48.3 |
| Context-PIPs (Ours) | 4 | 3 | 38.4 | 57.1 | 35.8 | 54.3 |
| Context-PIPs (Ours) | 4 | 6 | 41.0 | 58.3 | 36.2 | 54.8 |
| Context-PIPs (Ours) | 4 | 12 | 39.7 | 57.7 | 38.2 | 54.8 |
| Context-PIPs (Ours) | 6 | 12 | **42.7** | **60.3** | **40.2** | **57.0** |

## B  More Quantitative Comparisons

**PIPs Re-implementation**. There are two official PIPs [12] versions. PIPs (Paper) and PIPs (Released) respectively denote the model reported in the paper and the model provided in the released code. There are many misalignments between the paper description and the released code. We follow the parameters suggested in the paper and the released code but fail to reproduce the results. We, therefore, re-implement two PIPs as the baselines, according to the settings provided in the paper ($K = 6$) and the released code ($K = 4$). $K$ denotes the number of refinement iterations in training. The underscored PIPs (Re-imp.), i.e. $K = 6$, is the chosen baseline for comparison in the main paper. The performance of the re-implemented model is better than the numbers reported in the paper (Tab. A1). Although our re-implemented model presents inferior performance than the released model on FlyingThings++ and CroHD, they are comparable on TAP-Vid-DAVIS and the re-implement model is even better than the released model on TAP-Vid-Kinetics (Tab. A2). We add our proposed SOFE and TAFA modules to the re-implemented baselines to obtain our Context-PIPs models.

Table A3: Experiments on TAP-Vid-DAVIS (strided) and TAP-Vid-Kinetics (strided).

| Method | $K$ | MLP-Mixer Depth | TAP-Vid-DAVIS (strided) | | TAP-Vid-Kinetics (strided) | |
|---|---|---|---|---|---|---|
| | | | AJ | A-PCK | AJ | A-PCK |
| TAP-Net | - | - | 38.4 | 53.1 | 46.6 | 60.9 |
| PIPs (Re-imp.) | 4 | 6 | 41.0 | 56.6 | 37.2 | 55.5 |
| PIPs (Re-imp.) | 4 | 12 | 41.2 | 56.5 | 37.4 | 54.1 |
| PIPs (Re-imp.) | 4 | 16 | 41.4 | 56.8 | 38.6 | 55.3 |
| PIPs (Re-imp.) | 6 | 12 | 45.2 | 59.8 | 42.9 | 58.3 |
| PIPs (Released) | - | - | 45.6 | 60.6 | 39.6 | 55.6 |
| Context-PIPs (Ours) | 4 | 3 | 44.8 | 61.2 | 44.9 | 61.2 |
| Context-PIPs (Ours) | 4 | 6 | 45.9 | 61.9 | 45.4 | 62.1 |
| Context-PIPs (Ours) | 4 | 12 | 46.1 | 61.9 | 47.4 | 62.0 |
| Context-PIPs (Ours) | 6 | 12 | **48.9** | **64.0** | **49.8** | **64.3** |

We list the results for Flyingthings++ and CroHD in Tab. A1, the results for TAP-Vid-DAVIS (first) and TAP-Vid-Kinetics (first) in Tab. A2, and the results for TAP-Vid-DAVIS (strided) and TAP-Vid-Kinetics (strided) in Tab. A3. "first" and "strided" are two distinct query sampling strategies proposed by TAP-Vid [5], where "first" sampling only contains the initial visible query points, while "strided" sampling would contain all visible query points in every 5 frames.

The released PIPs model tends to overfit on FlyingThings++ because although it obtains the lowest error on FlyingThings++ but is inferior on TAP-Vid-DAVIS and TAP-Vid-Kinetics. Although we only show the Context-PIPs with $K = 4$ in the main paper, our $K = 6$ version achieves the best performance, outperforming PIPs $K = 6$ by 9.44% and 11.76% on DAVIS and Kinetics. Moreover, Context-PIPs trained with $K = 4$ and 3-layer MLP-Mixer achieves even better results than PIPs trained with $K = 6$ and 12-layer MLP-Mixer (Tab. A2).

