# OpenReview forum: "Context-PIPs: Persistent Independent Particles Demands Spatial Context Features"
_NeurIPS.cc/2023/Conference — NeurIPS 2023 spotlight_

### Official Review · Reviewer_2Jdq · 2023-07-05

**Soundness:** 3 good
**Presentation:** 2 fair
**Contribution:** 2 fair
**Rating:** 6
**Confidence:** 4

**Summary:**

This paper aims at estimating persistent long-term trajectories of query points in videos. For ignoring the potential benefits of 4 incorporating spatial context features, this paper argues that independent video point tracking also demands spatial context features. And this paper proposes a novel framework Context-TAP, which effectively improves point trajectory accuracy by aggregating spatial context features in videos. The framework Context-TAP contains two modules: 1) a source Feature Enhancement module, and 2) a target Feature Aggregation module, enhancing information with surrounding information from both source image and target image respectively. The Context-TAP ranks 1st on the four benchmarks 66 and shows clear performance superiority.

**Strengths:**

1). This paper is well-written and easy to understand
2). Experimental results are extensive and better than existing baseline models

**Weaknesses:**

The motivation and the solution to the problem are reasonable but seem regular approaches and it’s very common. The use of spatial context, regression samples to assist with features, has been seen very often. Therefore, the novelty of the

**Questions:**

1）In the SOFE module, are auxiliary features involved in the update iteration?
2）whether the inference process can be computed in parallel, and whether the computation time is the same or multiplied compared to the computation of only two frames of optical flow
3）Is it possible to visualize predicted samples to see if they actually capture semantically useful key points？


**Limitations:**

For the outlook in limitation is an important task to be solved in the TAP task, with a practical application scenario

---

> ### Author Rebuttal · Authors · 2023-08-08
>
> > The novelty of using spatial context.
>
> Spatial context is independent of temporal information. However, most of the existing methods mainly use temporal information.
>
> We argue that our spatial context feature component is novel:
>
> 1. We did not naively adopt common spatial context features techniques[1-2]. Instead, we fuse the cost information of the auxiliary context features for point tracking refinement, which has never been explored before. This may also further motivate the network design in other tasks.
>
> 2. We first show that target spatial context features improve point-tracking, which is ignored in closely related correspondence tasks such as optical flow and stereo matching.
>
> As suggested by Reviewer ujcD, our design may be further utilized in other tasks such as COTR and optical flow.
>
> > Are auxiliary features involved in the update iteration?
>
> Yes.
>
> > Whether the inference process can be computed in parallel
>
> Yes.
>
> > Whether the computation time is the same or multiplied compared to the computation of only two frames of optical flow.
>
> No. For an 8-frame Flyingthings++ sequence. The Context-TAP consumes 0.225s while 7 RAFT consumes 2.244s. Almost x10 slower.
>
> > Is it possible to visualize predicted samples to see if they actually capture semantically useful key points?
>
> Yes. We will add the visualization of learn-to-sample results in the final version.
>
> [1] Luo, Hao, et al. Object detection in video with spatial-temporal context aggregation.
>
> [2] Simon and Liu. Context-aware synthesis for video frame interpolation.

---

### Official Review · Reviewer_ujcD · 2023-07-05

**Soundness:** 3 good
**Presentation:** 3 good
**Contribution:** 4 excellent
**Rating:** 7
**Confidence:** 4

**Summary:**

This paper tackled the problem of Trakcing Any Point (TAP). Given a query point and a series of video frames, output all the coordinates corresponding to the query point in the video frame. The problem is interesting and can be regarded as an extension of optical flow.

PIPs only takes the features corresponding to the query point because this task estimate spoint trajectories independently. This paper takes the features surrounding the query point, called as ``spatial context features’’, to improve the accuracy of the point trajectory estimation. The motivation is reasonable. Experiments also demonstrate that Context-TAP outperformed PIPs by large margins and also improves the efficiency.

**Strengths:**

1. The proposed SOFE module learn to sample more features in the source image significantly improves the performance. I think the essence behind the module is the enlargement of perception field in the cost volume. In recent years, point-based correspondence estimation emerges, such as COTR[1] and ECO-TR[2]. The proposed technique may also be applied to this methods.
2. Context features are widely adopted in many SOTA correspondence models, such as stereo matching [3]and optical flow[4], i.e., the $F$ in Context-TAP, but they focus on the context features of the original pixel. This paper further aggregates the context features in the target images via attention mechanism (TAFA), which may also motivate the model design in the other correspondence tasks.
3. The motivation is well presented.
4. The experiments show that with the aid of the context features, the MLP-Mixer layers can be largely decreased, which improves the efficiency.
5. The ablation study is conducted thoroughly, revealing the necessity of the attention based TAFA and the number of samples.

[1] Jiang et. al. COTR: Correspondence Transformer for Matching Across Images
[2] Tan et. al. ECO-TR: Efficient Correspondences Finding Via Coarse-to-Fine Refinement
[3] Lipson et.al. RAFT-Stereo: Multilevel Recurrent Field Transforms for Stereo Matching
[4] Teed and Deng RAFT: Recurrent All-Pairs Field Transforms for Optical Flow



**Weaknesses:**

1.  TAP-Net also provide the occlusion accuracy (OA) metric. I am curious about how the spatial context features affect the occlusion prediction accuracy, but there is no discussion in the experiments. What is the reason?
2. The efficiency comparison only compares the number of parameters in the paper. I think time and memory usage is more important in practice.
3. I find that the authors provide a better model $K=6$ in the supplementary. What’s the reason? Should this model be moved to the main paper?
4. The benefit brought by the TAFA is not large.
5. The SOFE compute correlations between additional source features and target features. There are some other correlations that can be computed as context features. For example, the correlation between adjacent target images. Would these additional information also benefits TAP?


**Questions:**

Seeing Weakness.

**Limitations:**

Context-TAP adopts a fixed window size and handles long-term videos in a sliding manner. Context-TAP may struggle to track a point occluded in a long time interval.

---

> ### Author Rebuttal · Authors · 2023-08-08
>
> > TAP-Net also provide the occlusion accuracy (OA) metric. I am curious about how the spatial context features affect the occlusion prediction accuracy, but there is no discussion in the experiments. What is the reason?
>
> The OA comparison is listed below:
>
> | Method             | K | MLP-Mixer Depth | TAP-Vid-DAVIS (first) | TAP-Vid-Kinectics (first) | TAP-Vid-DAVIS (strided) | TAP-Vid-Kinetics (strided) |
> |--------------------|---|-----------------|-----------------------|---------------------------|-------------------------|----------------------------|
> | TAP-Net            | - | -               | 78.8                  | 80.6                      | 82.3                    | 85.0
> | PIPs (Re-imp.)     | 6 | 12              | 79.3                  | 77.0                      | 82.9                    | 81.5                       |
> | PIPs(Released)     | - | -               | 79.0                  | 75.7                      | 83.2                    | 81.0                       |
> | Context-TAP (Ours) | 6 | 12              | 79.5                  | 79.8                      | 83.4                    | 83.3                       |
>
> > The time and memory usage comparison.
>
> We will add the memory usage and FLOPs in the final version. When the number of MLP-Mixer layers is significantly reduced, the parameters of the network (11.54M v.s. 28.67M) and the FLOPs (216.4G v.s. 287.5G) evaluated with pytorch-OpCounter[1] for point tracking are reduced up to 59.7% and 24.7%.
>
> > Why the model K=6 is in the supplementary?
>
> Due to the limited time, we could not finish this experiment before the submission deadline. We will move the results to the main paper in the final version.
>
> > The benefit brought by the TAFA is not large.
>
> The improvement is still large by reducing the 5.9% Average Trajectory Error of Occluded Points (ATE-Occ) on CroHD.
>
> > Would more additional information also benefit TAP?
>
> Thanks for your suggestions. We believe so and this will become our future work.
>
> [1] https://github.com/Lyken17/pytorch-OpCounter

---

> > ### Comment · Reviewer_ujcD · 2023-08-19
> > **Reply authors**
> >
> > Thanks for the authors' responses. I still hold on my score.

---

### Official Review · Reviewer_WGNK · 2023-07-07

**Soundness:** 3 good
**Presentation:** 3 good
**Contribution:** 3 good
**Rating:** 6
**Confidence:** 4

**Summary:**

This paper presents some technical improvements to PIPs, which is a state-of-the-art method for multi-frame pixel tracking. There are two key modifications to the architecture. The first is: rather than only use the feature of the target to represent the appearance of the target, look at the cost map and estimate a few offsets, and sample additional appearance features at these offsets; this yields additional correlations to be used by the temporal model (the MLP-Mixer). The second modification is: on each frame, do some QKV attention to obtain some additional features, and use these to perform a feature update, in addition to (or instead of (this wasn't entirely clear)) the feature update normally done by the model. These modifications yield 1-2 points gain in accuracy in each dataset considered.

**Strengths:**

Using additional appearance information from the first frame, and additional appearance information across time, are very sensible contributions over PIPs. It is also exciting that these modifications were achieved without increasing the computational complexity of the method.


**Weaknesses:**

For me, the main weaknesses have to do with clarity, mostly in the writing, but also in the results.

In the experiments, it would be nice to see the "d_avg" metric reported on the TAP-Vid benchmarks, as computed in the TAP-Vid paper. Or maybe this is re-named here to A-PCK? Overall, in Table 1 and Table 2, it does not seem like the PIPs results on DAVIS or Kinetics exactly match the table from the TAP-Vid paper. Why is this?

There is some difference between "PIPs (Paper)" and "PIPs (Re-imp.)", but I could not find text talking about this. I understand that the PIPs github provides a model slightly improved over the original paper, plus maybe some bug-fixes. Is that the re-implementation referenced here? Or is this a new implementation, produced independently by the authors?

"We train our Context-TAP and PIPs with different MLP-Mixer depths, i.e., the number of layers in the MLP-Mixer, to show the extraordinary efficiency and effectiveness of our proposed Context-TAP."

I am not sure that training with different MLP-mixer depths has anything to do with "extraordinary" aspects of the method or results. Perhaps this sentence should be re-considered, or supported in some way?



Overall, considering that the method here builds on PIPs and not TAP-Net, it seems to me that a less confusing name would be something like Context-PIPs (rather than Context-TAP).

I think the acronyms SOFE and TAFA are not improving the clarity of the paper.

In lines 62-67, the first contribution seems to be just a high-level idea; the second contribution seems to be two independent contributions, and the third contribution is actually a measurement of the middle contributions. I think this could be rewritten into two good contributions plus an evaluation.

"PIPs and TAP solve the video particle tracking problem in a similar manner, i.e., recurrently refining multi-frame point trajectory via correlation maps."

It seems not accurate to say that TAP-Net involves iterative/recurrent refinement.


SOurse -> Source


eveluated -> evaluated





**Questions:**

"The augmented correlation features Cˆ in Eq. 3 encode abundant visual similarities. Therefore, we generate a query from it to extract target context features ..."

I do not understand this part. The correlations contain similarity information, but not appearance information. It sounds like they are being used to create an appearance query. How (or why) does this work?


**Limitations:**

Looks fine

---

> ### Author Rebuttal · Authors · 2023-08-08
>
> > In the experiments, it would be nice to see the "d_avg" metric reported on the TAP-Vid benchmarks, as computed in the TAP-Vid paper. Or maybe this is re-named here to A-PCK?
>
> Yes. We just renamed "d_avg" to A-PCK here because the Average Percentage of Correct Keypoints (PCK) is a more common terminology in correspondence tasks[1-3]. We will clarify the name in the revised version.
>
> > Overall, in Table 1 and Table 2, it does not seem like the PIPs results on DAVIS or Kinetics exactly match the table from the TAP-Vid paper. Why is this?
>
> Because we cannot exactly reimplement the performance of the released model even with the official repository. We thus provide three versions of PIPs: PIPs (paper), PIPs (re-implement), and PIPs (released). We align all the training settings of PIPs (re-implement) to our Context-TAP for a fair comparison. The other two results are provided in the supplementary for reference. Notice that our PIPs (re-implement) is even better than PIPs (released) on Kinetics because the released PIPs tends to overfit FlyingThings++ dataset, as discussed in L12 of our supplementary.
>
> TAP-Net evaluates PIPs on the DAVIS and Kinectics with the released model. We also provide such model named "PIPs (released)". Its performance is improved because we slightly tune the visibility threshold in the chaining rule of PIPs. Specifically, for a video longer than 8 frames, PIPs iteratively selects new visible starting points and chains the trajectory. In PIPs, a point is visible if the predicted visibility is larger than 0.9. However, we observe that there is a significant visibility distribution bias on the training set of FlyingThings++. We use 0.9 as the visibility threshold on DAVIS so that PIPs achieved slightly better results than those reported by TAP-Net.
>
> > There is some difference between "PIPs (Paper)" and "PIPs (Re-imp.)", but I could not find text talking about this. I understand that the PIPs github provides a model slightly improved over the original paper, plus maybe some bug-fixes. Is that the re-implementation referenced here? Or is this a new implementation, produced independently by the authors?
>
> We provide the details of our PIPs re-implementation in the L12 of our supplementary materials and will make it clearer in the final version. We use the code from the PIPs repository to re-implement the PIPs. As stated in the previous question, we cannot perfectly reproduce the numbers reported by the PIPs paper or the model released in the repository, so we align the training settings of PIPs with our Context-TAP for a fair comparison.
>
> > I am not sure that training with different MLP-mixer depths has anything to do with "extraordinary" aspects of the method or results. Perhaps this sentence should be re-considered, or supported in some way?
>
> Here we want to express that when the spatial context information is introduced, comparable results can be obtained even if the number of MLP-Mixer layers is significantly reduced. The parameters of the network (11.54M v.s. 28.67M) and the FLOPs (216.4G v.s. 287.5G) evaluated with pytorch-OpCounter[4] for point tracking are reduced up to 59.7% and 24.7%, which is prominent. We will replace the extraordinary with "prominent".
>
> > Overall, considering that the method here builds on PIPs and not TAP-Net, it seems to me that a less confusing name would be something like Context-PIPs (rather than Context-TAP).
>
> A5.Thanks for your suggestion. The main reason we use TAP is that we think Tracking Any Point (TAP) is a more general and easy-to-understand term compared with Persistent Independent Particles (PIPs).
>
> > I think the acronyms SOFE and TAFA are not improving the clarity of the paper.
>
> The acronyms SOFE and TAFA are mainly to shorten the name of the module, which is a commonly adopted strategy in previous literatures such as LoFTR[1], RAFT[6], and GMA[5]. We will also carefully consider the naming in the final version, such as Source feature Enhancement Module (SEM) and Target feature Aggregation Module (TAM).
>
> > In lines 62-67, I think this could be rewritten into two good contributions plus an evaluation.
>
> Thanks for the suggestion! We will reorganize the description of our contribution.
>
> >"PIPs and TAP solve the video particle tracking problem in a similar manner, i.e., recurrently refining multi-frame point trajectory via correlation maps."
> > It seems not accurate to say that TAP-Net involves iterative/recurrent refinement.
> > SOurse -> Source
> > eveluated -> evaluated
>
> We would refine the statement and typos in the final version.
>
> > "The augmented correlation features Cˆ in Eq. 3 encode abundant visual similarities. Therefore, we generate a query from it to extract target context features ..."
> > I do not understand this part. The correlations contain similarity information, but not appearance information. It sounds like they are being used to create an appearance query. How (or why) does this work?
>
> Pixels that hold similar appearance features share similar motions, so that we learn to find auxiliary features that provide informative motion cues through the feature similarities, i.e., correlations. This design is inspired by GMA [5], which smoothes the optical flow estimation via averaging the flow with appearance similarities.
>
> [1] Sun, Jiaming, et al. LoFTR: Detector-free local feature matching with transformers.
>
> [2] Truong, Prune, et al. Pdc-net+: Enhanced probabilistic dense correspondence network.
>
> [3] Huang, Zhaoyang, et al. Neuralmarker: A framework for learning general marker correspondence.
>
> [4] https://github.com/Lyken17/pytorch-OpCounter
>
> [5] Jiang, Shihao, et al. Learning to estimate hidden motions with global motion aggregation.
>
> [6] Zachary and Deng. Raft: Recurrent all-pairs field transforms for optical flow.

---

### Official Review · Reviewer_ydfF · 2023-07-12

**Soundness:** 2 fair
**Presentation:** 2 fair
**Contribution:** 2 fair
**Rating:** 6
**Confidence:** 2

**Summary:**

This work proposes a method for video point tracking. It is built upon the previous method of persistent independent particles (PIPs). The authors add context features to the source and target feature encoding in PIPs. The resulting method is called Context-TAP.

The proposed method is evaluated on multiple benchmarks for video point tracking. Improved tracking accuracy is observed over PIPs and other baseline methods.

**Strengths:**

+ Using more context features is tried and true in many vision tasks. This work shows it is also helping with the PIPs method.

+ The paper is generally well written with thorough experiments. The authors provide a detailed ablation study on the design parameters in this method.

**Weaknesses:**

- My primary concern is about the scope of this work. The claim is that video point tracking needs contextual information, but the execution is a modification to a specific method for this task. Whether the proposed modification is general enough to boost multiple methods or only specific to PIPs is unknown.


**Questions:**

- Based on my primary concern, my first question is: do the proposed modules work on any other method that deals with this task?


- From the comparison with baseline methods, the numeric improvement of tracking accuracy seems less drastic than that of PIPs over other methods. An explanation of the accuracy improvement would help justify the significance of this work.

**Limitations:**

The authors have stated the limitations of this work.

---

> ### Author Rebuttal · Authors · 2023-08-08
>
> > Do the proposed modules work on any other method that deals with this task?
>
> Yes, we think so because both recent point-tracking networks[1-2] are built upon PIPs, which iteratively refine the point trajectories via cost information. However, they only take the features of the query points into consideration. It is natural to add our modules to gain benefit from the spatial context. We only test our modules on PIPs because it is the first and only method that provides open-sourced PyTorch code for point-tracking. Applying our modules to other point-tracking networks would be our future work.
>
> > From the comparison with baseline methods, the numeric improvement of tracking accuracy seems less drastic than that of PIPs over other methods. An explanation of the accuracy improvement would help justify the significance of this work.
>
> Our Context-TAP outperforms PIPs by reducing 11.4% Average Trajectory Error of Occluded Points (ATE-Occ) on CroHD and increasing 11.8% Average Percentage of Correct Keypoint (A-PCK) on TAP-Vid-Kinectics. PIPs’ improvement is drastic because it is the first method that learns point tracking with longer temporal information and the new FlyingThings++ dataset. The methods which PIPs compared to, such as RAFT, were not designed for the TAP task.
>
> [1] Carl et.al. TAPIR: Tracking Any Point with per-frame Initialization and temporal Refinement.
>
> [2] Yang, et al. PointOdyssey: A Large-Scale Synthetic Dataset for Long-Term Point Tracking.

---

### Decision · Program_Chairs · 2023-09-21

**Decision:**

Accept (spotlight)

**Comment:**

All reviewers voted to accept the paper. The authors have promised to include metrics not reported along with runtime details in the final camera ready. The AC also agrees with the reviewer that the title is confusing. Given the paper extends the PIP method and not the TAP method the AC highly recommends changing the title to reflect this (perhaps Context-PIP) as Context-TAP strongly suggests the method is extending TAP. With these changes the AC has determined that this paper should be accepted to Neurips 2023.